# Sex differences in the fecal microbiome and hippocampal glial morphology following diet and antibiotic treatment

Anju Saxena[1], Roberta R. M. Moran[1], Meghan R. Bullard[2], Emma O. Bondy[1], Matthew Foster Smith[1], Lainie Morris[2], Nicaella Fogle[1], Jagroop Singh[2], Brendan Jarvis[1], Tammy Ray[1], Juhi Saxena[1], Linnea Ruth Freeman[1,2]*

1 Neurosciences, Furman University, Greenville, South Carolina, United States of America, 2 Department of Biology, Furman University, Greenville, South Carolina, United States of America

* linnea.freeman@furman.edu

**Data Availability Statement:** I have uploaded the data to FigShare. It is now published: https://doi.org/10.6084/m9.figshare.14935665

## Abstract

Rising obesity rates have become a major public health concern within the United States. Understanding the systemic and neural effects of obesity is crucial in designing preventive and therapeutic measures. In previous studies, administration of a high fat diet has induced significant weight gain for mouse models of obesity. Interestingly, sex differences in high-fat diet-induced weight gain have been observed, with female mice gaining significantly less weight compared to male mice on the same high-fat diet. It has also been observed that consumption of a high-fat diet can increase neurogliosis, but the mechanism by which this occurs is still not fully understood. Recent research has suggested that the gut microbiome may mediate diet-induced glial activation. The current study aimed to (1) analyze changes to the gut microbiome following consumption of a high fat (HF) diet as well as antibiotic treatment, (2) evaluate hippocampal microgliosis and astrogliosis, and (3) identify sex differences within these responses. We administered a low fat (Research Diets D12450 K) or high fat diet (Research Diets D12451) to male and female C57Bl/6 mice for sixteen weeks. Mice received an antibiotic cocktail containing 0.5g/L of vancomycin, 1.0 g/L ampicillin, 1.0 g/L neomycin, and 1.0 g/L metronidazole in their drinking water during the last six weeks of the study and were compared to control mice receiving normal drinking water throughout the study. We observed a significant reduction in gut microbiome diversity for groups that received the antibiotic cocktail, as determined by Illumina next-generation sequencing. Male mice fed the HF diet (± antibiotics) had significantly greater body weights compared to all other groups. And, female mice fed the low fat (LF) diet and administered antibiotics revealed significantly decreased microgliosis and astrogliosis in the hippocampus compared to LF-fed females without antibiotics. Interestingly, male mice fed the LF diet and administered antibiotics revealed significantly increased microgliosis, but decreased astrogliosis, compared to LF-fed males without antibiotics. The observed sex differences in LF-fed mice given antibiotics brings forward questions about sex differences in nutrient metabolism, gut microbiome composition, and response to antibiotics.

**Funding:** Research reported in this publication was supported by the National Institute Of General Medical Sciences of the National Institutes of Health under Award Number P20GM103499 (LRF). The content is solely the responsibility of the authors and does not necessarily represent the official views of the National Institutes of Health.

**Competing interests:** The authors have declared that no competing interests exist.

## Introduction

Rising obesity rates have become a major public health concern within the United States. By 2030, the obesity prevalence in the United States is expected to rise to an alarming 46–47% [1]. Obesity rates in 1999–2000 were 30.9% for US adults [2] and 42.4% in 2017–2018 [3], indicating the considerable rise in obesity prevalence during the last twenty years. Understanding the systemic and neural effects of obesity is crucial in designing preventive and therapeutic measures.

Obesity is a complex metabolic disease characterized by excessive weight gain and a body mass index greater than 30 kg/m2 [4]. Co-morbidities such as diabetes, cancer, and cardiovascular disease lead to increased mortality [5, 6]. Consumption of a Western Diet is a contributing factor to the rising obesity rates. The Western Diet refers to a high fat diet consisting of high levels of saturated fats and simple sugars [7, 8].

In previous studies, administration of a high fat diet has induced significant weight gain for mouse models of obesity [8–11]. Interestingly, sex differences in high fat diet-induced weight gain have been observed; female mice gain significantly less weight compared to male mice on the same high fat diet [10, 12]. These sex differences extend to adipose tissue inflammation, hyperinsulinemia, and islet hypertrophy, with female mice demonstrating a resistance to these conditions compared to male mice [13]. While estrogen, cytokine, and gut metabolite levels have all been implicated as potential mediators of these observed sex differences, the mechanism by which these sex differences arise is still not well understood [13–16].

In addition to weight gain, consumption of a high fat diet alters gut microbiome composition [8, 16]. The gut microbiome refers to the microorganisms, and their genetic material, that reside within the gut. In response to a high-fat diet, the gut microbiome undergoes compositional changes that can decrease the diversity of bacteria and thereby induce gut dysbiosis [9, 17]. High fat diets have been shown to change the bacterial levels of the two most abundant phyla of bacteria within the gut, increasing Firmicutes and decreasing Bacteroidetes [5, 18, 19]. Changes within these bacterial phyla can also alter levels of bacterial metabolites such as lipopolysaccharides and short-chain fatty acids (SCFAs), which contribute to systemic effects of obesity [20–23].

Furthermore, high fat diet consumption can activate and alter the morphology and function of glial cells within the brain, including microglia and astrocytes [9, 11, 24–27]. Microglial cells are the resident immune cells of the brain and exhibit changes in number and activation state in response to inflammatory events [28, 29]. Astrocytes have a number of roles in the brain, including metabolic, immune, and vascular support; they are the most abundant glial cell in the brain [11, 30, 31]. The hippocampus, a brain region involved in learning and memory, is vulnerable to obesity-induced inflammation. High saturated fat and refined sugar intake, characteristic of the Western Diet, are linked to deficits in hippocampal-dependent learning and memory in adults [32, 33]. For example, obese mice perform significantly worse than controls on tasks that require hippocampal cognitive function [34–36]. Cognitive impairment in diet-induced obesity is associated with increased microglial activation [37]. An increase in activated microglia has been reported in the hippocampus of mice fed a high fat diet [38, 39] and a high fat, high sugar diet [9].

While it has already been established that consumption of a high fat diet can induce glial activation, the mechanism by which this occurs is still not fully understood. Recent research has suggested that the gut microbiome may mediate this effect [17]. Consumption of a high fat diet can induce compositional changes within the gut microbiome and these compositional changes could drive the neuroinflammatory response within the brain. There is a growing

body of evidence that suggests the gut microbiome and its metabolites may mediate the neuroinflammatory response to a high-fat diet.

In order to investigate the role of gut microbiome diversity on hippocampal microglia morphology, we included administration of an antibiotic cocktail in the drinking water to significantly deplete gut microbiome diversity and test its effects. Specifically, we measured the fecal microbiome as an indicator of changes occurring at the level of the gut microbiome. We included the antibiotics vancomycin, ampicillin, neomycin, and metronidazole as has been previously performed in mice and resulted in microbiota depletion [40–42]. The current study aimed to (1) analyze changes to the gut microbiome following consumption of a high fat (HF) diet as well as antibiotic treatment, (2) evaluate hippocampal microgliosis and astrogliosis, and (3) identify sex differences within these responses. Given previous results that have shown sex differences in high fat diet-induced obesity, gut microbiome composition, as well as neurogliosis, we designed this study in order to evaluate each of these components. The antibiotic cocktail was administered in order to significantly deplete gut microbiome composition and diversity and therefore, test the effects when this variable was altered. We hypothesized that severely depleting gut microbiome diversity in both sexes would normalize differences in neurogliosis. If sex differences in the gut microbiome are a factor for sex differences in neurogliosis, then significantly depleting gut microbiome diversity in both sexes should lead to similar changes to neuroglia.

## Materials and methods

### Animals, diets, and treatments

Sixty-five C57Bl/6 mice (The Jackson Laboratory, Bar Harbor, ME, USA) were housed 2–4 to a cage under a controlled, 12-hour light/ 12-hour dark cycle with *ad libitum* access to food and water. Mice were received at 6 weeks and were allowed to acclimate to the vivarium for 2 weeks. Two-month-old male (n = 32) and female (n = 33) mice were randomly assigned to the following diet groups: High Fat (HF) (n = 33) and Low Fat (LF) (n = 32). Mice were then further divided into antibiotic groups: antibiotic drinking water (n = 35) and normal drinking water (n = 30). This created a total of eight experimental groups: Male LF (n = 6), Male LF Antibiotics (n = 9), Male HF (n = 7), Male HF Antibiotics (n = 10), Female LF (n = 9), Female LF Antibiotics (n = 8), Female HF (n = 8), and Female HF Antibiotics (n = 8).

The LF control diet consisted of (by calorie): 20% protein, 70% carbohydrate, and 10% fat (D12450 K; Research Diets Inc. New Brunswick, NJ, USA). The HF treatment diet consisted of (by calorie): 20% protein, 35% carbohydrate, and 45% fat (D12451; Research Diets Inc., New Brunswick, NJ, USA). Soybean oil and lard were the primary sources of fat for these diets. Both diets had the same amount of vitamin and mineral content. The antibiotic cocktail was administered through the drinking water and consisted of 0.5g/L of vancomycin, 1.0 g/L ampicillin, 1.0 g/L neomycin, and 1.0 g/L metronidazole (Cayman Chemical, Ann Arbor, MI, USA). The diets were administered for 16 weeks. Antibiotics were added to the drinking water during the last 6 weeks of the total 16-week study. Throughout the 16-week study, body weight and food consumption were evaluated weekly by manually weighing the mice and the food remaining in the home cage. Caloric consumption was calculated by multiplying the food consumption for the cage for the LF diet by 3.82 kcals/g and 4.7 kcals/g for the HF diet. These values were further divided by the number of animals in the cage to determine the estimated caloric consumption for individual animals. This was done in order to compare caloric consumption across cages with varying numbers of animals (2-4/cage); it was not intended to obtain accurate individual consumption. Animal protocols were approved by the Furman University Animal Care and Use Committee; we received written consent for the protocol

(Approval Number 13-18-02). Our protocols were carried out according to the guidelines from the National Institute of Health.

## Microbiome analyses

Fecal samples were collected using sterile technique on the last day of the study (after 16 weeks of diet administration, at the time of euthanization) and stored in microcentrifuge tubes at -80˚C until analysis. Genomic DNA was isolated from fecal samples using the PowerSoil® DNA Isolation Kit (Qiagen) following the manufacturer's instructions. As an alternative to the recommended 250mg of soil, approximately 200mg of fecal sample was added to the Power-Beads tube to undergo cell lysis. The purified DNA was eluted from the spin filter using 50uL of solution C6 and stored at -20˚C until PCR amplification. The 16S universal Eubacterial primers `515F GTGCCAGCMGCCGCGGTAA` and `806R GGACTACVSGGGTATCTAAT` were utilized to evaluate the microbial ecology of each sample on the HiSeq 2500 with methods via the bTEFAP® DNA analysis service. Each sample underwent a single-step 30 cycle PCR using HotStarTaq Plus Master Mix Kit (Qiagen, Valencia, CA, USA) and were used under the following conditions: 94˚C for 3 minutes, followed by 30 cycles of 94˚C for 30 seconds; 53˚C for 40 seconds and 72˚C for 1 minute; after which a final elongation step at 72˚C for 5 minutes was performed. Following PCR, all amplicon products from different samples were mixed in equal concentrations and purified using Agencourt Ampure beads (Agencourt Bioscience Corporation, MA, USA). Samples were sequenced utilizing the Illumina MiSeq chemistry following manufacturer's protocols.

The Q25 sequence data derived from the sequencing process was processed using a proprietary analysis pipeline (www.mrdnalab.com, MR DNA, Shallowater, TX). Sequences were depleted of barcodes and primers, then short sequences < 200bp were removed; sequences with ambiguous base calls were removed, and sequences with homopolymer runs exceeding 6bp removed. Sequences were then denoised and chimeras removed. Operational taxonomic units were defined after removal of singleton sequences, and clustering at 3% divergence (97% similarity) [43–46]. OTUs were then taxonomically classified using BLASTn against a curated NCBI database and compiled into each taxonomic level.

Alpha and beta diversity analysis was conducted as described previously [43–46] using Qiime 2 [47]. Significance reported for any analysis is defined as $p < 0.05$.

## Tissue collection and immunohistochemistry

Mice were anesthetized deeply with isoflurane gas and then decapitated. Brain collection was conducted immediately following decapitation. The right hemisphere of the brain was drop-fixed with 4% paraformaldehyde for 48 hours, and then cryoprotected in 30% sucrose in 0.1 M phosphate buffered saline at 4˚C. Coronal sections (40 μm) through the dorsal hippocampus were collected using a cryostat (Microm HM 505e). Sections were then processed using standard immunohistochemistry procedures as described by previously published protocols [9, 48]. Every 6th brain section through the dorsal hippocampus was evaluated. First, sections were incubated with ionized calcium binding adaptor protein (Iba-1, 1:1,000, FUJIFILM Wako, Osaka, Japan) or glial fibrillary acidic protein (GFAP; 1:1,000, Abcam, Cambridge, MA, USA) at 4˚C overnight. After primary antibody incubation, the tissue was washed in phosphate buffered saline with Triton X-100 (detergent) and then incubated in biotinylated goat anti-rabbit secondary antibody (1:500, Jackson ImmunoResearch, West Grove, PA, USA) for 2 hours at room temperature. Sections were washed and placed in Elite ABC reagent (Vector Laboratories, Burlingame, CA, USA) for 1.5 hours. The tissue was developed using a Vector VIP Kit (Vector Laboratories, Burlingame, CA, USA). Finally, sections were washed, mounted, and

coverslipped on slides with Vectamount (Vector Laboratories, Burlingame, CA, USA). Slides were then analyzed and photographed using a light microscope.

## Analysis of microglial cells

Microglial cells rapidly change morphology in response to immune challenges in the local environment; they can change states between ramified/resting to activated and phagocytic/amoeboid. The changes occur along a spectrum rather than a binary switch [28]. Therefore, characterizing features of microglia using FracLac and Skeleton analyses allows for comparisons between groups [29, 49]. Overall, resting microglial cells are characterized by long processes and extensive branching while activated microglial cells have a dense cell body and minimal branching [28, 50]. To evaluate these states, microglia in the hippocampus were evaluated based on branch length plus endpoints (FracLac), density, span ratio (cell shape), circularity, fractal dimension (cell complexity), and endpoints (branch number). Microglia were imaged at 100x and processed using ImageJ (National Institute of Health, https://imagej.nih.gov/ij/). Both the Analyze Skeleton Plugin (http://imagej.net/AnalyzeSkeleton31) and the FracLac for ImageJ Plugin (http://rsb.info.nih.gov/ij/plugins/frac-lac.html) were used to analyze the microglial cells, as performed previously [9, 29, 49]. The Analyze Skeleton Plugin allowed for the determination of the number of summed microglial endpoints per soma and the number of summed microglial branch lengths per soma. Six to seven images, each containing at least one microglial cell, from each animal (3–5 per group), were captured in the dorsal hippocampus. The researcher capturing images was blind to the groups. A different researcher, also blind to the groups, performed ImageJ analyses. Each image had the FFT Bandpass filter applied to the image, the brightness adjusted, and the unsharp mask and despeckle tools were also used to refine the image before thresholding. After thresholding the image, it was again despeckled, made binary, and the outliers were removed. The background noise of the image around the microglial cell was then cleared manually by the researcher. The image was then converted to a skeleton form using the Analyze Skeleton Plugin and skeletonize features. This allowed for the collection of tag skeleton features that were relevant to determining the microglial activation state: slab voxels (branch length) and endpoints. Values for branch length less than 1.75 mm were removed and the FracLac Plugin was used to determine: endpoint plus branch length, cell complexity (fractal dimension), density, circularity, and cell shape (span ratio).

## Analysis of astrocytes

Microscopy images were captured at 10x and then processed using ImageJ (National Institute of Health, https://imagej.nih.gov/ij/). With ImageJ, a threshold was applied and then the percentage of area covered by astrocytes in each image was calculated, as performed previously [9]. 3–4 images per mouse were analyzed from 4–5 animals per treatment group. Data were analyzed by two different researchers, blind to the groups, and their results were averaged.

## Statistical analysis

All statistical tests were conducted using GraphPad Prism (GraphPad Software, La Jolla, California, USA, www. graphpad.com). For food consumption, a mixed-effects analysis (sex x treatment x time) was performed due to a few missing values (researcher error). Body weights were log transformed and then analyzed using a three-way ANOVA (sex x treatment x time). And, a two-way (sex x treatment) ANOVA was used to determine significant differences in microglia and astrocyte measurements (data were normally distributed according to the Shapiro-Wilk test). A Tukey's posthoc analysis was conducted for any measures that indicated

statistically significant differences for the ANOVA. Fecal microbiome results were analyzed using a Kruskal–Wallis test followed by a two-stage linear step-up procedure of Benjamini, Krieger, and Yekutieli in order to control for multiple comparisons by controlling the false discovery rate. Fecal microbiome data were analyzed for all eight groups, and a separate analysis excluding the antibiotic-treated groups was also conducted in order to evaluate sex and diet effects, alone.

## Results

### Food consumption and body weights

A significant effect of time ($F_{(3.320, 184.4)}$ = 86.86; p <0.0001), a significant effect of treatment ($F_{(7, 57)}$ = 47.99; p <0.0001), and a significant interaction effect for time and treatment ($F_{(105, 833)}$ = 5.959; p <0.0001) were observed for caloric consumption over the whole study period (data not shown). Significant differences between average caloric consumption for the whole study were determined between Male LF and Female LF (p = 0.0018), Male LF and Male LF Antibiotics (p<0.0001), Male LF Antibiotics and Male HF Antibiotics (p<0.0001), Female LF Antibiotics and Female HF Antibiotics (p<0.0001), Female HF and Female HF Antibiotics (p = 0.0023), Male HF Antibiotics and Female HF Antibiotics (p<0.0001) (Fig 1A).

There was a significant effect of treatment ($F_{(3, 57)}$ = 14.39; p<0.0001) and a significant effect of sex ($F_{(1, 57)}$ = 15.76; p = 0.0002) on body weight gain (Fig 1B). There was also a significant interaction effect between sex and treatment ($F_{(3, 57)}$ = 5.822; p = 0.0015) on body weight gain. Specifically, Male HF gained significantly more weight than Male LF (p = 0.0005). Additionally, Male HF gained significantly more weight than Female HF (p = 0.0032), demonstrating sex differences in weight gain. Male HF Antibiotics gained significantly more weight than Female HF Antibiotics (p = 0.0052) and Male LF Antibiotics (p = 0.0004).

### Microbiome analyses

After stringent quality sequence curation, a total of 2,656,509 sequences were parsed and 2,534,448 were then clustered. 2,486,339 sequences identified within the Bacteria and Archaea domains were utilized for final microbiota analyses. The average reads per sample was 51,798. For alpha and beta diversity analysis, samples were rarefied to 20,000 sequences.

We determined a significant reduction in diversity indices (Shannon Diversity; Fig 2A, Operational Taxonomic Unit; Fig 2B, Faith's Phylogenetic Diversity; Fig 2C) due to antibiotic treatment, with some variation due to diet and/or sex. Interestingly, Female LF Antibiotics revealed an increased Shannon Diversity index compared to all other antibiotic-treated groups (compared to Male LF Antibiotics, p = 0.0096; compared to Female HF Antibiotics, p = 0.0248; compared to Male HF Antibiotics, p = 0.0131; Fig 2A).

The principal coordinate (PCoA) plot of weighted UniFrac data further displays differences between antibiotic and normal drinking water groups (Fig 3). Interestingly, Male LF Antibiotics (pink), Male HF Antibiotics (yellow) and Female HF Antibiotics (blue) reveal very similar results between animals, within their groups. On the other hand, Female LF Antibiotics (green) reveals greater variance in results between animals, which is also found for normal drinking water groups. The double dendrogram (Fig 4) revealed a significant effect of antibiotic treatment on gut microbiome diversity. Furthermore, there is some additional clustering based on diet.

Fecal microbiome genera abundances revealed differences due to sex, diet, and/or antibiotic treatment (Fig 5). For male and female mice fed the LF and HF diet, antibiotics significantly depleted most genera of bacteria. *Lactococcus* was the dominant genus of bacteria for antibiotic-treated groups.

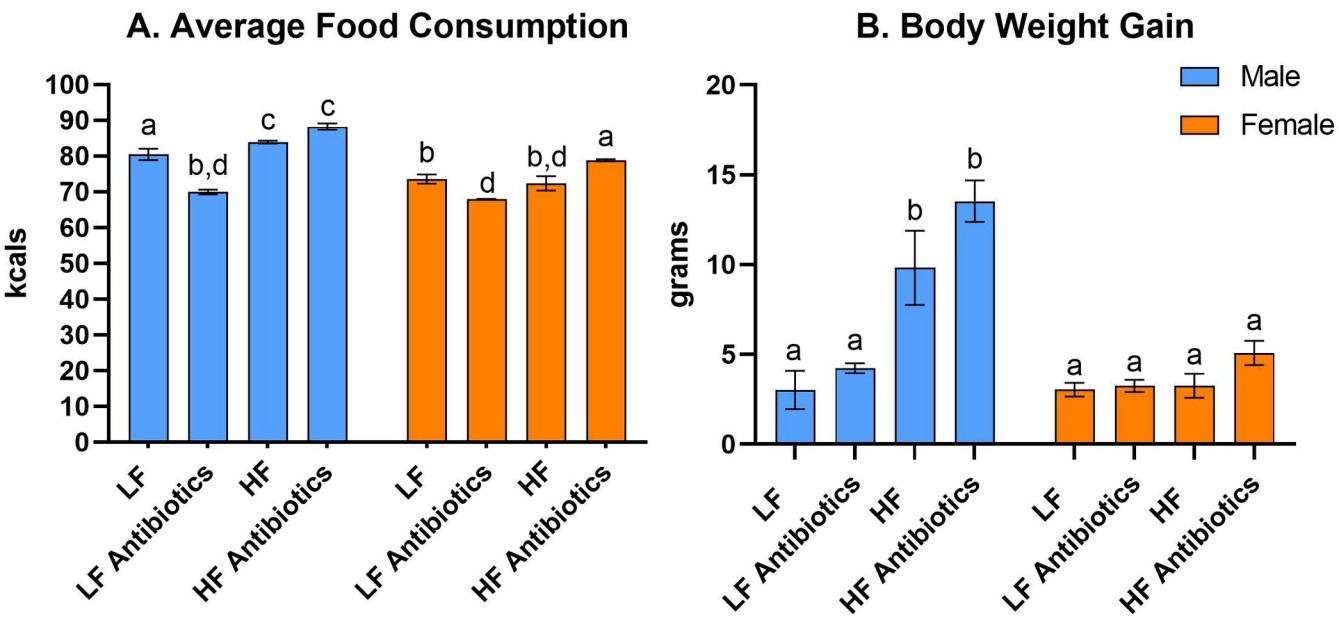

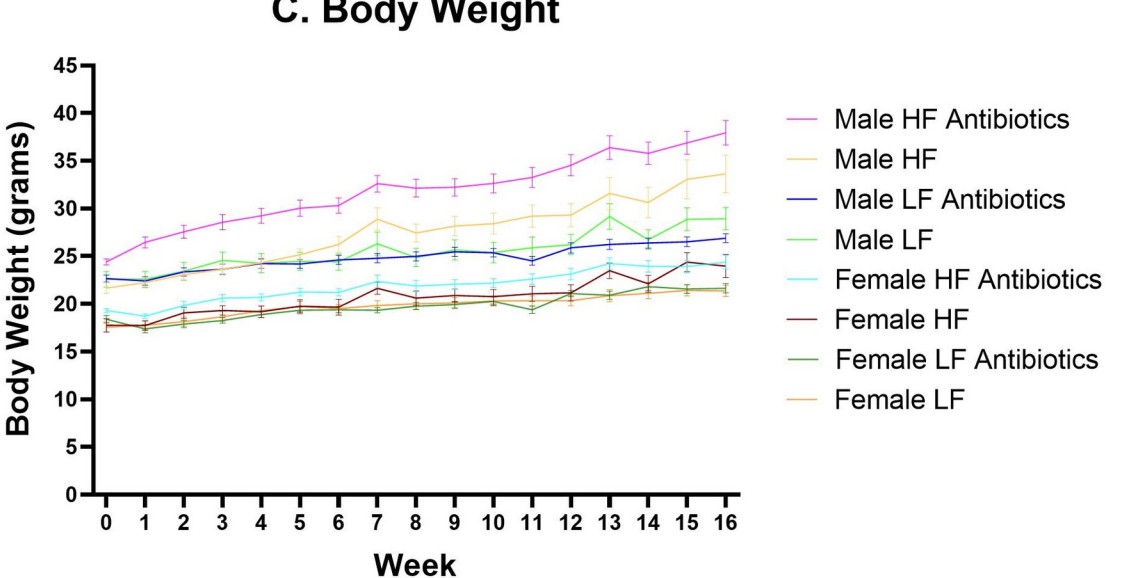

**Fig 1. Average food consumption and body weight during the 16-week dietary study.** (A) Average food consumption for the entire 16-week study. Different letters indicate groups that are significantly different (p<0.05). (B) HF-fed mice gained significantly more weight than LF-fed mice. Male HF and Male HF Antibiotics gained significantly more weight than all other groups. (C) Male HF Antibiotics reveal the highest body weight; Female LF and Female LF Antibiotics reveal the lowest body weight. Values shown are mean ± SEM.

We determined statistically significant differences for fecal microbiome genera due to antibiotics, as well as sex and diet alone (Table 1). The HF diet significantly changed genera abundances for both sexes compared to the LF diet. For example, *Clostridium*, *Lachnoclostridium*, *Oscillospira*, *Sporobacter*, and *Tyzzerella* were increased to a similar extent for both male and

## A. Shannon Diversity

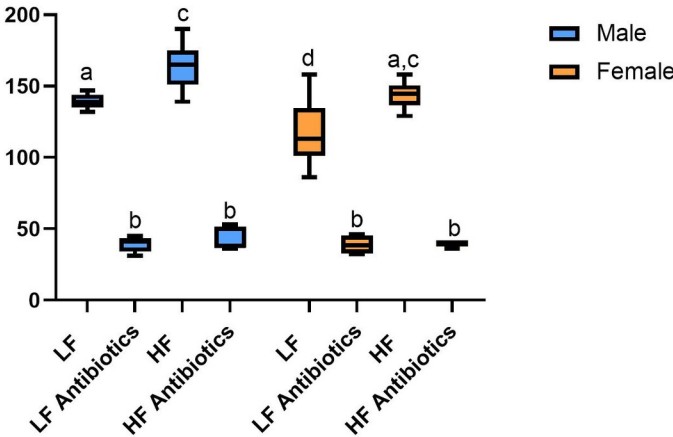

## B. Operational Taxonomic Unit

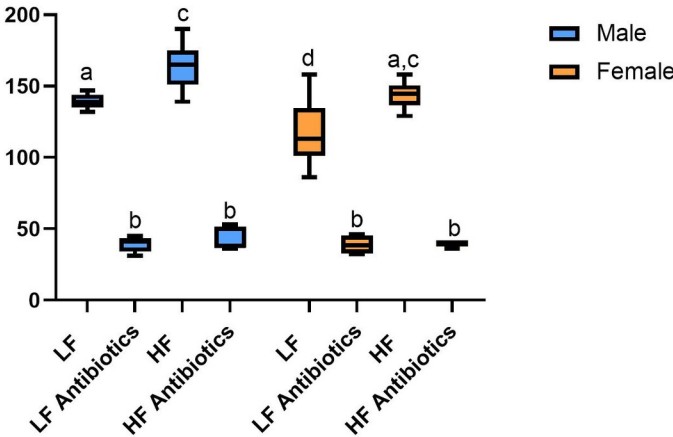

## C. Faith's Phylogenetic Diversity

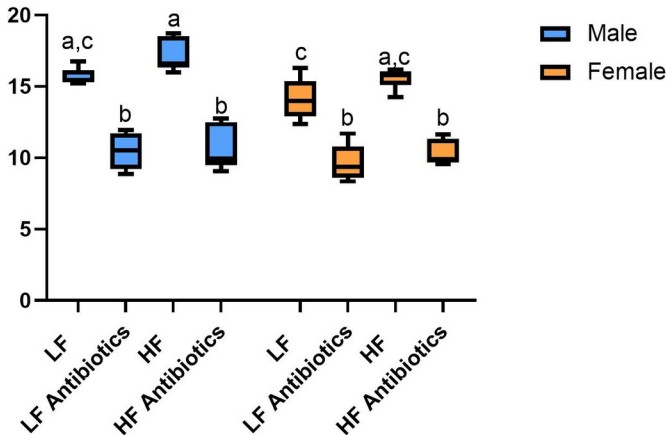

**Fig 2. Fecal microbiome alpha diversity (n = 6/group).** (A) There was a significant effect of treatment (p<0.0001) and a significant interaction effect (sex x treatment; p = 0.0002) for Shannon Diversity. (B) There was a significant effect of treatment (p<0.0001) and a significant effect of sex (p = 0.0021) on Operational Taxonomic Units. (C) There

was a significant effect of treatment (p<0.0001) and a significant effect of sex (p = 0.0026) on Faith's Phylogenetic Diversity.

female mice fed the HF diet compared to males and females fed the LF diet. *Allobaculum* and *Citrobacter* were decreased to a similar extent for male and female HF mice compared to male and female LF mice. On the other hand, *Lactobacillus* was significantly higher in Female LF mice compared to all other groups. *Turicibacter* and *Blautia* were significantly lower in Female HF mice compared to all other groups. Lastly, *Barnesiella* was significantly higher in Female HF mice compared to all other groups.

## Microglial morphology characterization

FracLac and Skeleton analyses were performed to evaluate microgliosis. There was a significant sex x treatment interaction (F(3,27) = 15.02, p < 0.0001), significant effect of treatment (F(3,27) = 5.547, p = 0.0042), and significant effect of sex (F(1,27) = 7.073, p = 0.0130) for fractal analysis (Fig 6A). Female LF Antibiotics revealed significantly greater values for endpoint and branch length per soma compared to Female LF (p = 0.0002) and Male LF Antibiotics (p<0.0001). Interestingly, Male LF Antibiotics displayed decreased fractal analysis values compared to Male LF (p = 0.0432). The increased values for Female LF Antibiotics are indicative of resting microglia, with numerous, long processes.

There was a significant effect of treatment (F(3,27) = 5.633, p = 0.0039), sex (F(1,27) = 6.751, p = 0.0150) and a significant sex x treatment interaction (F(3,27) = 14.44, p < 0.0001) on branch length (Fig 6B), supporting the findings for fractal analysis: Female LF Antibiotics revealed the greatest branch length compared to the other groups. A significant sex x treatment interaction was also determined for cell complexity (F(3, 27) = 3.753, p = 0.0225; Fig 6C). There was a significant effect of treatment on cell shape (F(3,27) = 3.133, p = 0.0419; Fig 6D). Lastly, there was a significant effect of treatment (F(3,27) = 4.367, p = 0.0125) and a significant interaction effect (F(3,27) = 11.65, p < 0.0001) for summed endpoints (Fig 6F). Altogether, these data point to microglia in a resting-like state for Female LF Antibiotics and an activated-

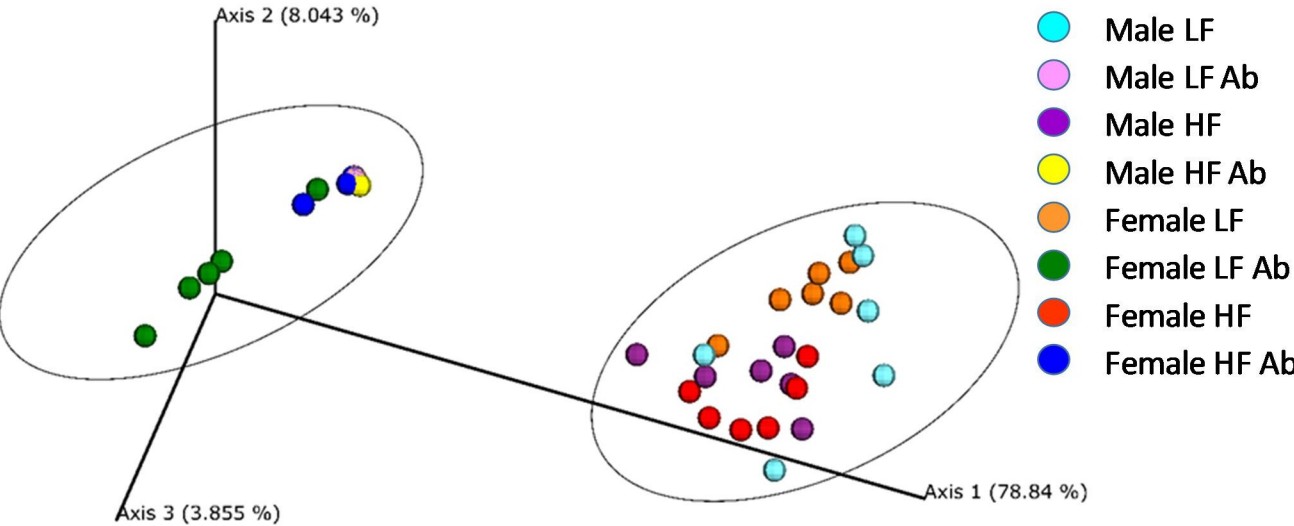

**Fig 3. Principle coordinate plot of weighted UniFrac data (n = 6/group).** The primary vector explains 78.8% of the variation between the groups. The first 3 vectors together exhibit 90.7% of the variation among the groups.

# Double Dendrogram

## Groups

**Fig 4. Dual hierarchical dendrogram (n = 6/group).** To provide a visual overview combined with analysis, a dual hierarchal dendrogram was used to display the data for the predominant genera with clustering related to the different groups. Samples with more similar microbial populations were mathematically clustered closer together. The genera (consortium) were used for clustering. The samples with more similar consortium of genera cluster closer together with the length of connecting lines (top of heatmap) related to the similarity, shorter lines between two samples indicate closely-matched microbial consortium. The heatmap represents the relative percentages of each genus. The predominant genera are represented along the right Y-axis. The legend for the heatmap is provided in the upper left corner. The heat map and clustering reveals a significant effect of antibiotic treatment.

like state for Male LF Antibiotics. Antibiotics did not significantly impact HF-fed mice, regardless of sex. However, Female HF Antibiotics did reveal more endpoints, greater branch length, and greater cell complexity compared to Female HF. Male HF Antibiotics displayed slightly greater endpoints, branch length, and complexity compared to Male HF. Representative micrographs of microglia taken at 100X are displayed in Fig 7.

Fractal analysis correlated with the Shannon Diversity index. Interestingly, there seems to be a positive and negative correlation, depending on which group is plotted. In Fig 8A, data

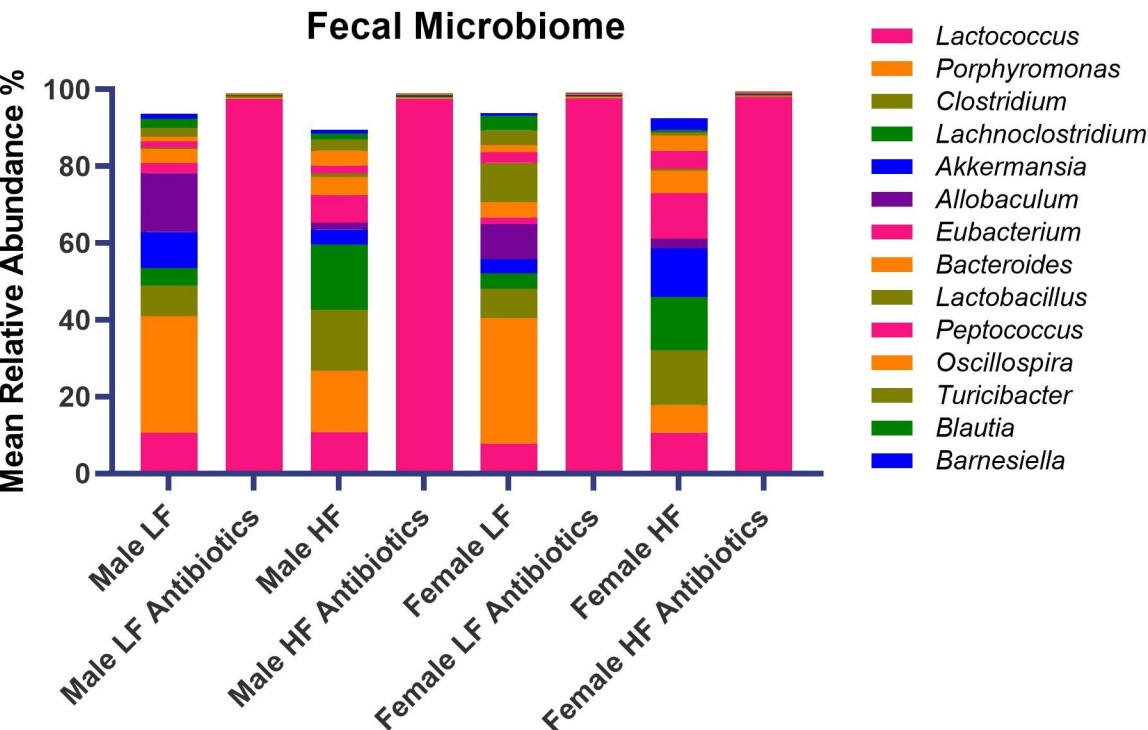

**Fig 5. Genus-level relative abundances in the fecal microbiome (n = 6/group).** Average relative abundance for the top 14 most abundant genera are displayed for each of the 8 experimental groups.

from all eight groups is presented. Antibiotic-treated groups revealed a low Shannon Diversity index and a positive correlation with Fractal Analysis. On the other hand, normal drinking water-treated groups revealed a higher Shannon Diversity index and a negative correlation with Fractal analysis. When analyzed separately, antibiotic-treated groups reveal a significant positive correlation (Spearman r = 0.5637; p = 0.0203; Fig 8B). Normal drinking water groups revealed a significant negative correlation (Spearman r = -0.5735; p = 0.0223; Fig 8C).

## Astrocyte analysis

There was a significant treatment effect for GFAP densitometry (F(3,31) = 13.18, p<0.0001; Fig 9). Overall, Male and Female LF Antibiotics revealed decreased GFAP density compared to the other groups, reaching statistical significance in the Female LF Antibiotics group compared to the Female LF group (p = 0.0007). Representative micrographs of astrocytes taken at 10X in the CA1 region of the hippocampus are presented.

## Discussion

Previous research has reported increased weight gain in male mice compared to female mice, despite being fed the same obesity-inducing diet [9, 10]. In this study, male mice gained significantly greater weight compared to female mice. Proposed explanations for this female resistance to weight gain include increased estrogen levels, lower levels of the pro-inflammatory cytokine IL-6, and increased levels of the short-chain fatty acid, butyrate (a gut metabolite) compared to male mice [13–16]. However, further research is needed to fully understand the mechanism by which resistance to diet-induced obesity occurs in female mice. Additionally, our study demonstrated that the Male HF Antibiotics group had the highest average weight

**Table 1. Statistically significant differences in gut microbiome relative abundances.**

| Genus | Male LF vs. Male LF Antibiotics | Male HF vs. Male HF Antibiotics | Female LF vs. Female LF Antibiotics | Female HF vs. Female HF Antibiotics | Male LF vs. Male HF | Female LF vs. Female HF | Male LF vs. Female LF | Male HF vs. Female HF |
|---|---|---|---|---|---|---|---|---|
| Lactococcus | p = 0.0246 | p = 0.0015 | p = 0.0023 | p = 0.0007 | | | | |
| Porphyromonas | p = 0.0003 | p = 0.0133 | p = 0.0002 | p = 0.0373 | | p = 0.0043 | | |
| Clostridium | p = 0.0069 | p = 0.0007 | p = 0.0100 | p = 0.0013 | p = 0.0338 | p = 0.0338 | | |
| Lachnoclostridium | p = 0.0320 | p = 0.0010 | p = 0.0078 | p = 0.0002 | p = 0.0029 | p = 0.0062 | | |
| Akkermansia | p = 0.0008 | p = 0.0141 | p = 0.0209 | p = 0.0003 | | | | |
| Allobaculum | p = 0.0028 | | p = 0.0003 | p = 0.0100 | p = 0.0019 | p = 0.0412 | | |
| Eubacterium | p = 0.0061 | p = 0.0021 | | p < 0.0001 | p = 0.0455 | p = 0.0002 | | |
| Bacteroides | p = 0.0039 | p = 0.0020 | p = 0.0069 | p = 0.0014 | | | | |
| Lactobacillus | | p = 0.0133 | p = 0.0003 | p = 0.0246 | | p = 0.0114 | p < 0.0001 | |
| Peptococcus | p = 0.0069 | | p = 0.0003 | p = 0.0011 | | | | |
| Oscillospira | p = 0.0069 | p = 0.0023 | p = 0.0094 | p = 0.0004 | p = 0.0055 | p = 0.0179 | | |
| Turicibacter | p = 0.0088 | p = 0.0065 | p = 0.0002 | p = 0.0221 | | p = 0.0007 | | p = 0.0247 |
| Blautia | p = 0.0005 | p = 0.0036 | p = 0.0015 | p = 0.0198 | | p = 0.0004 | | p = 0.0455 |
| Barnesiella | p = 0.0015 | | p = 0.0141 | p < 0.0001 | | p = 0.0008 | | p = 0.0062 |
| Parasutterella | p = 0.0094 | p = 0.0106 | p = 0.0011 | p = 0.0005 | | | | |
| Bifidobacterium | | | p = 0.0004 | p = 0.0150 | | | | |
| Ruminococcus | p = 0.0209 | p < 0.0001 | p = 0.0112 | p = 0.0047 | p = 0.0005 | | | |
| Sporobacter | p = 0.0187 | p = 0.0002 | p = 0.0392 | p = 0.0001 | p = 0.0029 | p = 0.0062 | | |
| Citrobacter | p = 0.0002 | p = 0.0233 | p = 0.0003 | p = 0.0337 | p = 0.0043 | p = 0.0025 | | |
| Candidatus soleaferrea | p = 0.0233 | p = 0.0007 | p = 0.0074 | p = 0.0004 | p = 0.0019 | | | |
| Tyzzerella | p = 0.0158 | p = 0.0009 | p = 0.0119 | p = 0.0003 | p = 0.0017 | p = 0.0062 | | |
| Pelotomaculum | p = 0.0412 | p = 0.0177 | p = 0.0014 | p < 0.0001 | | | | |
| Streptococcus | p = 0.0001 | p = 0.0119 | p = 0.0057 | p = 0.0078 | | | | |
| Roseburia | p = 0.0073 | p < 0.0001 | | p = 0.0004 | p = 0.0007 | p = 0.0128 | | |
| Pseudoflavonifractor | p = 0.0150 | p = 0.0001 | p = 0.0304 | p = 0.0006 | p = 0.0076 | | | |
| Dorea | p = 0.0369 | p < 0.0001 | p = 0.0249 | p = 0.0006 | p = 0.0037 | p = 0.0048 | | |
| Paludibacter | p = 0.0043 | p = 0.0010 | p = 0.0084 | p = 0.0024 | | | | |
| Spirochaeta | p = 0.0013 | p = 0.0049 | p = 0.0001 | p = 0.0439 | | | | |
| Adlercreutzia | p = 0.0012 | p = 0.0019 | p = 0.0045 | p = 0.0058 | | | | |

gain. This aligns with research that suggests an association between antibiotic usage and increased weight gain, with early antibiotic exposure resulting in the greatest amount of weight gain [51–53]. Furthermore, antibiotic usage is known to increase animal growth and weight and is a common practice for livestock farming [54]. Despite its known effect on animals, further research into prolonged antibiotic exposure and its relationship with adult human weight gain is needed, especially due to the rising use of antibiotics in modern society.

Caloric consumption for individual animals was calculated and recorded each week during the 16-week study. Since the animals were housed 2 to 4 in a cage, individual caloric consumption for each animal was calculated by dividing the total number of calories consumed for the entire cage by the number of mice within the cage. While this does present a limitation to our study as we did not accurately determine individual mouse caloric consumption, individual housing of mice was not preferred as it has been shown to significantly impact weight gain and food consumption [55]. Fortunately, all mice were able to remain in group-housing conditions throughout the entire study. Regardless of sex, antibiotic treatment lowered caloric intake for

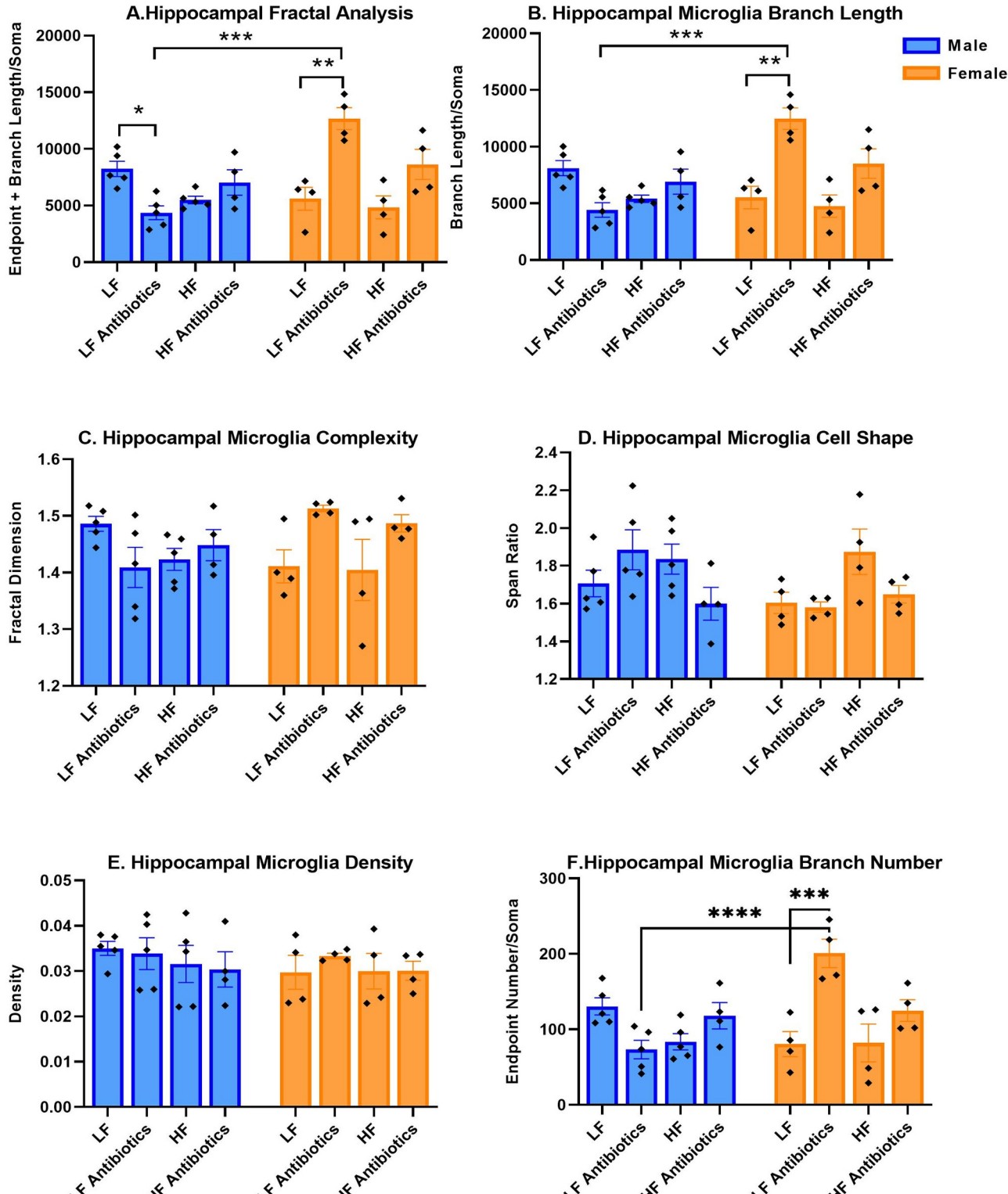

**Fig 6. Microglial analysis (n = 4-5/group).** The results for fractal analysis (A), branch length (B), cell complexity (C), cell shape (D), density (E), and summed endpoints (F) are shown. * p<0.05; *** p<0.001; **** p<0.0001.

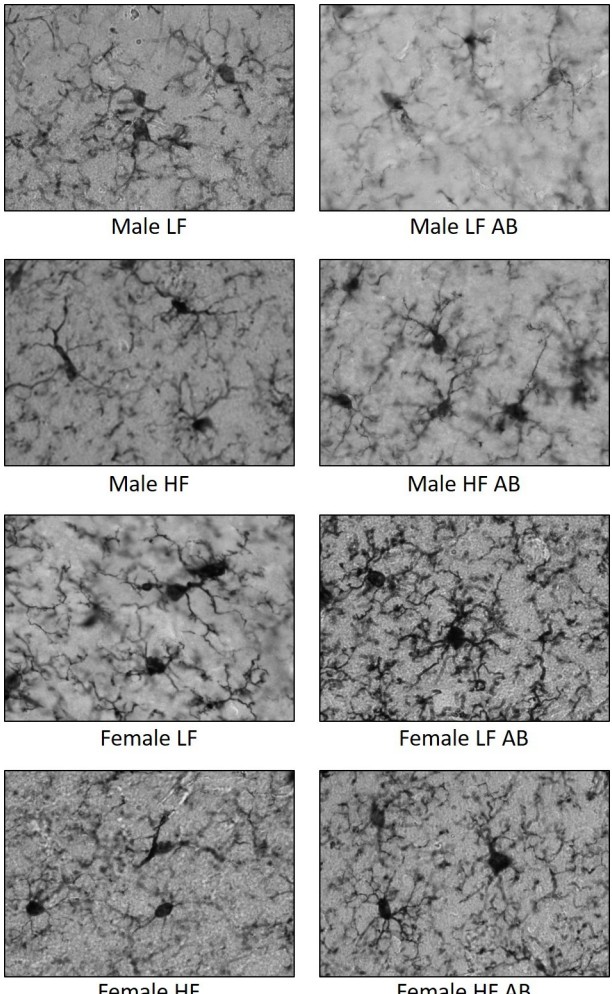

**Fig 7. Representative micrographs of microglia.** 100X images were collected for microglia labeled with the Iba-1 antibody.

LF groups and increased caloric intake for HF groups. Male HF and Male HF Antibiotics consumed, on average, significantly more calories than Female HF and Female HF Antibiotics, respectively. Male LF consumed significantly more calories than Female LF. However, there were no significant differences in caloric consumption between Male LF Antibiotics and Female LF Antibiotics.

Sex differences and treatment effects were also determined for the fecal microbiome. In a previous study, we reported genus-level differences due to sex and/or diet [9]. Interestingly, we determined different effects to fecal bacteria in our current study. For example, we previously determined significant differences due to sex for *Allobaculum*, *Peptococcus*, *Blautia*, *Ruminococcus*, *Pelotomaculum*, *Paludibacter*, and *Adlercreutzia* that we did not determine in the current study. In the current study, we determined that *Lactobacillus*, *Turicibacter*, *Blautia*, *and Barnesiella* were altered due to sex. The housing conditions, diet, and source for the mice were the same between studies. However, it has previously been shown that stochastic changes to the microbiome can occur despite controlling multiple variables including housing conditions, animal vendor, and diet [56]. We did determine some overlapping results between our previous and current study: *Allobaculum*, *Eubacterium*, and *Citrobacter* changed due to diet.

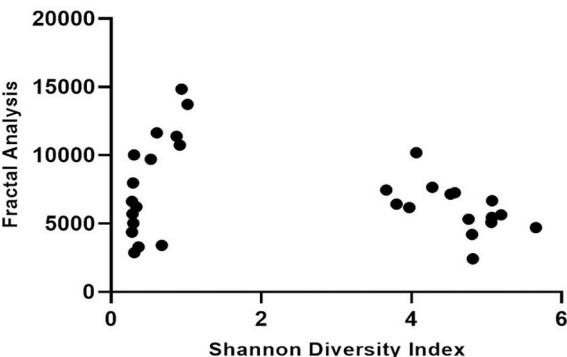

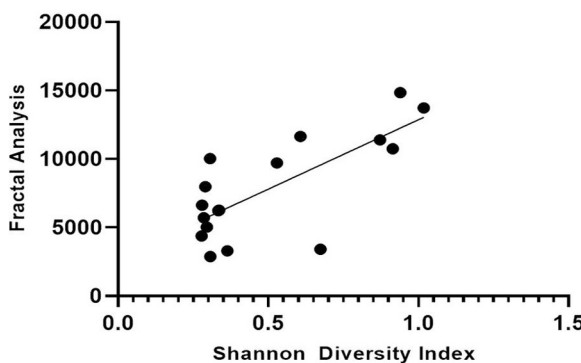

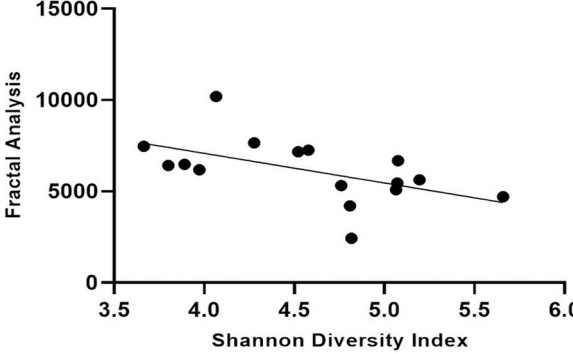

**Fig 8. Correlations for fractal analysis and Shannon Diversity.** (A) All groups are plotted, including results with low Shannon diversity for antibiotic-treated vs. higher Shannon diversity for mice that received normal drinking water. (B) A positive correlation (Spearman r = 0.5637; p = 0.0203) was determined for antibiotic-treated mice. (C) A negative correlation (Spearman r = -0.5735; p = 0.0223) was determined for normal drinking water mice.

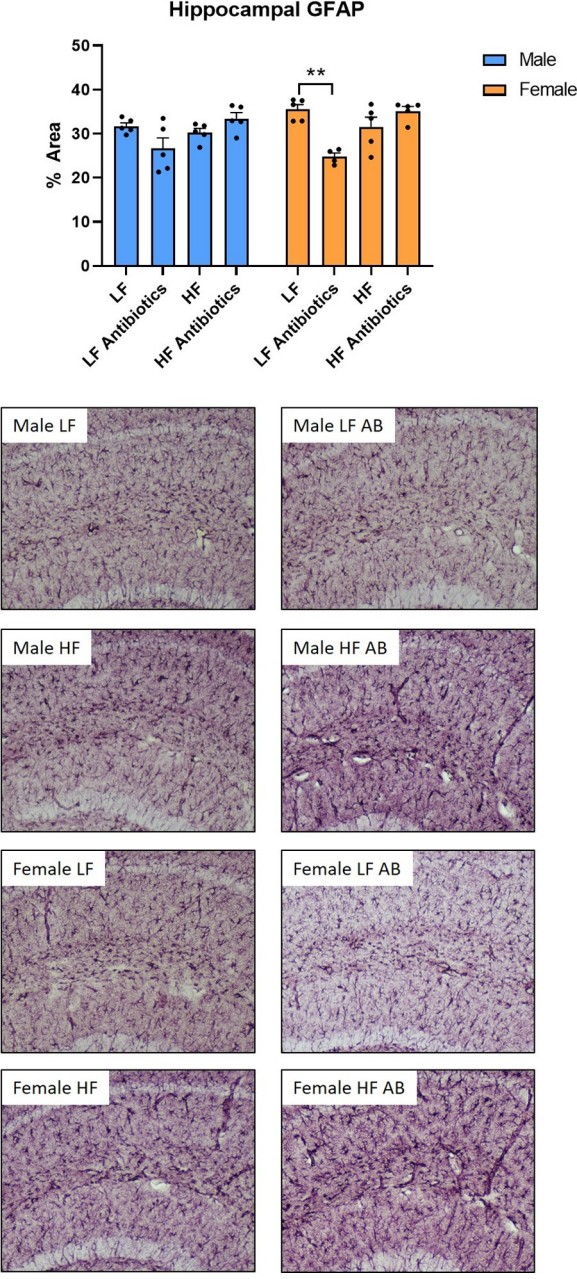

**Fig 9. GFAP densitometry (n = 5/group).** Female LF Antibiotics revealed a significant decrease in GFAP densitometry in the hippocampus compared to Female LF. ** $p < 0.01$. Representative micrographs of images taken at 10X following labeling with a GFAP antibody are presented.

Antibiotic-treated mice revealed a significant reduction in diversity as well as significant depletion of specific genera of bacteria. *Lactococcus* was the primary genus of bacteria present in antibiotic-treated mice, regardless of sex. For all bacteria shown in Fig 5, antibiotics decreased the abundance, except for *Lactococcus*, which was significantly greater in abundance compared to control animals given normal drinking water. It is possible that without competition from other genera of bacteria, *Lactococcus* could over-populate the microbiome. Interestingly, Female LF Antibiotics revealed significantly higher Shannon diversity compared to the other

antibiotic-treated groups. Female LF Antibiotics also lacked a significant change to *Eubacterium* and *Roseburia* abundance compared to Female LF, whereas all other groups indicated significant decreases in these genera when compared to their control counterparts. It is not likely that these differences fully account for the observed differences in neurogliosis, however it is possible that gut microbiome diversity contributes to altered glial health [57, 58]. We did determine that Shannon diversity correlated with fractal analysis (Frac Lac; Fig 8). Previous studies have shown that increased species richness within the gut microbiome is correlated with decreased microgliosis [57, 59].

Previous studies have also shown that males have greater amounts of microglial cells compared to females. However, females have a greater number of activated microglial cells [60]. In the present study, we characterized hippocampal microglial cell features using fractal analysis (branch length plus endpoints per soma), fractal dimension (cell complexity), circularity, density, and span ratio (cell shape) values. Activated microglial cells are characterized by decreased values for branch length plus endpoints, cell complexity (fractal dimension), and increased values for span ratio, circularity, and density. In contrast, extensive branching and greater cell complexity are indicative of resting microglial cells. We observed features of resting microglia for Female LF Antibiotics, including increased branch length, increased branch number, and increased cell complexity (not statistically significant). On the other hand, Male LF Antibiotics revealed a significantly lower value for fractal analysis compared to Male LF. This change could indicate more activated microglia for male LF-fed mice following antibiotics administration. We were surprised that antibiotic administration led to striking sex differences, and only in LF-fed mice.

In a recent study by Cordella et al., male mice fed a standard diet and administered gentamicin and vancomycin in drinking water supplemented with sugar revealed increased microglial density, but no change to morphology, in the hippocampus [61]. This study only focused on male mice and therefore, sex differences under the experimental parameters were not assessed. However, a lack of change to microglial morphology following antibiotic administration is in contrast to our findings. It is possible that since our low fat diet does not contain sucrose, it caused additional changes to microglia morphology, and possibly, their metabolism. We did not supplement the drinking water with sucrose. In a study by Zarrinpar et al., male mice were administered an antibiotic cocktail with the same antibiotics that we utilized, but also included the anti-fungal, amphotericin B. Both the control group and antibiotic-treatment group were fed a normal-chow diet (LabDiet 5001). They determined that antibiotic treatment improved glucose tolerance and insulin sensitivity [62], as also shown in previous studies [42, 63, 64]. It was further reported that antibiotic treatment increased levels of glucagon-like peptide 1 (GLP-1) as well as decreased the short chain fatty acid (SCFA), butyrate. Given increased energy demand in the colon due to higher levels of GLP-1 and the lack of butyrate as fuel, the gut could increase glucose utilization, in particular, via anaerobic glycolysis. This would result in cecal enterocytes utilizing higher levels of glucose, leading to overall changes to metabolic homeostasis throughout the body. In our study, the low fat diet did not deliver sucrose and we did not include any sucrose in the drinking water. The main source of carbohydrate for the diet was corn starch. While corn starch consists of glucose polymers, it is possible that there is a reduced capacity for breakdown of starch in the intestine following antibiotic administration and subsequently, decreased absorption of glucose. In a future experiment, we will need to evaluate blood glucose levels and insulin sensitivity. Interestingly, it has been proposed that glucose metabolism exerts transcriptional control over microglial activation [65–67]. Glucose is the primary fuel for microglia, as it is for neurons. Under inflammatory conditions, glucose transporter 1 (GLUT1) is upregulated in order for glucose uptake and glycolysis to increase in microglia. When glucose is unavailable, free fatty acids can be a secondary fuel for microglia

[68]. It is possible that in our study, the low-fat, no sucrose diet in addition to a depleted microbiome, resulted in a lack of fuel in the form of glucose, as well as the back-up fuel in the form of free fatty acids. This lack of fuel could have inhibited microglial activation in Female LF Antibiotics mice. We did not observe the same effect in male mice, which could be protected by their slightly higher adiposity. However, Male LF Antibiotics did not consume more calories than Female LF Antibiotics. It is also possible that baseline sex differences in glucose and lipid metabolism as well as gut microbiome composition impacted these results. In a recent study by Peng et al., both male and female C57Bl/6 mice were fed either a chow diet (Research Diets D12450B) or high fat diet (Research Diets D12451) and administered the same antibiotic cocktail as in our study. However, antibiotics were provided before diet administration and the chow diet animals were not included for antibiotic treatment. Their study reported increased weight gain, insulin resistance, and fasting blood glucose in male mice compared to female mice. Additionally, they determined sex differences in the fecal microbiome. For example, *Parabacteroides*, *Lactobacillus*, *and Bifidobacterium* were increased in female mice compared to male mice [69]. We also observed significantly higher levels of *Lactobacillus* for our Female LF mice, but this was significantly decreased following antibiotic treatment. Previous research has determined that some strains of *Lactobacillus* have anti-inflammatory effects [69, 70]. Peng et al. also observed decreased *Roseburia* in high fat-fed male mice, but it was increased in mice administered antibiotics. On the other hand, *Roseburia* was decreased in antibiotic-treated high fat-fed female mice. In our study, high fat diet treatment significantly increased *Roseburia* abundance for both sexes, but to a greater extent in male mice. Antibiotic treatment significantly decreased *Roseburia* abundance in both sexes and both diets. In the study by Peng et al., *Roseburia* was implicated in improved glucose metabolism and it was further discussed that *Roseburia* is a proposed probiotic that increases production of SCFAs [69]. The difference in abundance of this genus of bacteria between the two studies may be due to timing of antibiotic treatment; we included antibiotic treatment for 6 weeks following 10 weeks of diet as well as in conjunction with diet, while Peng et al. pre-treated with antibiotics for 4 weeks, stopped antibiotic treatment, and then started the high fat diet. We determined sex differences in fecal microbiome composition, which could result in altered metabolite production, inflammation, and overall metabolism.

Furthermore, metronidazole was included in our antibiotic cocktail. Metronidazole is an absorbable antibiotic that does cross the blood brain barrier (BBB), which could lead to differences in its direct effects on microglial activation. However, the extent to which metronidazole penetrates the BBB and the subsequent effect on microglia is not yet fully understood [50, 71]. Future experiments, including those investigating sex differences in response to metronidazole, should be conducted. Additionally, Female LF Antibiotics revealed decreased GFAP expression compared to Female LF. Male LF Antibiotics also revealed decreased GFAP expression compared to Male LF, but not to the same extent as females. Astrocytes are involved in numerous functions, from immune support to BBB support to metabolism. However, it is possible that the decrease in GFAP expression for LF-fed, antibiotic-administered mice is an indicator of decreased nutrient load and decreased metabolic activity. The LF diet is lower in fat, does not contain sucrose, and is lower in calories; and, the decreased bacterial load following antibiotics administration could further contribute to a decrease in nutrients due to altered microbial metabolism.

The limitations of our study include a lack of metabolic measurements, including fasting glucose, insulin sensitivity, and SCFA analysis, a lack of analysis for the contribution of metronidazole and its direct effects on the brain, and a lack of analysis of the gut-brain axis. Here, we focused on sex differences in the fecal microbiome and hippocampal microglia characterization. An alternative approach could include testing germ-free mice, however, changes to

microglial morphology have already been determined with the germ-free mouse model [43] and our goal was to deplete the microbiome short-term following exposure to the low fat or high fat diet.Next steps will require further investigation of metabolic parameters as well as directly probing the gut-brain axis in order to better understand its role. Nonetheless, we report sex differences in fecal microbiome composition as well as morphological differences in hippocampal morphology in response to diet and antibiotic treatment.

## Conclusions

We report sex, diet, and antibiotic effects on weight gain, food consumption, and fecal microbiome composition. We further report significantly decreased microgliosis and astrogliosis in the hippocampus for Female LF Antibiotics compared to Female LF whereas Male LF Antibiotics revealed increased microgliosis compared to Male LF. We hypothesize this effect is due to reduced fuel and subsequently, reduced capacity for microglial activation in Female LF Antibiotics mice. This sex difference brings forward questions about sex differences in nutrient metabolism, gut microbiome composition, response to antibiotics, and neurogliosis.

Future studies investigating the direct contribution of diet and the direct contribution of antibiotics to neuroglial health are necessary. Furthermore, the pathway(s) along the gut-brain axis that modulate glial response, including their sex differences, are an essential next-step.

## Author Contributions

**Conceptualization:** Linnea Ruth Freeman.

**Data curation:** Anju Saxena, Roberta R. M. Moran, Meghan R. Bullard, Emma O. Bondy, Matthew Foster Smith, Lainie Morris, Nicaella Fogle, Jagroop Singh, Brendan Jarvis, Tammy Ray, Juhi Saxena.

**Formal analysis:** Anju Saxena, Roberta R. M. Moran, Matthew Foster Smith, Lainie Morris, Jagroop Singh, Juhi Saxena, Linnea Ruth Freeman.

**Funding acquisition:** Linnea Ruth Freeman.

**Investigation:** Anju Saxena, Roberta R. M. Moran.

**Project administration:** Linnea Ruth Freeman.

**Resources:** Linnea Ruth Freeman.

**Supervision:** Linnea Ruth Freeman.

**Validation:** Linnea Ruth Freeman.

**Writing – original draft:** Anju Saxena, Roberta R. M. Moran, Matthew Foster Smith, Linnea Ruth Freeman.

**Writing – review & editing:** Linnea Ruth Freeman.

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
