## [Decision Letter · Decision Letter 0]

19 Oct 2021

PONE-D-21-28243Sex differences in the gut microbiome and hippocampal glial morphology following diet and antibiotic treatmentPLOS ONE

Dear Dr. Freeman,

Thank you for submitting your manuscript to PLOS ONE. After careful consideration, we feel that it has merit but does not fully meet PLOS ONE’s publication criteria as it currently stands. Therefore, we invite you to submit a revised version of the manuscript that addresses the points raised during the review process.

We look forward to receiving your revised manuscript.

Kind regards,

Christopher Staley, Ph.D.

Academic Editor

PLOS ONE

Additional Editor Comments (if provided):

While the reviewers found the topic to be of interest, clarifications are recommended throughout by both reviewers. In addition, the link to the gut-brain axis must be made clear or the text modified accordingly.

Journal Requirements:

Reviewers' comments:

Reviewer's Responses to Questions

**Comments to the Author**

1. Is the manuscript technically sound, and do the data support the conclusions?

Reviewer #1: Yes

Reviewer #2: Yes

2. Has the statistical analysis been performed appropriately and rigorously? 

Reviewer #1: No

Reviewer #2: Yes

3. Have the authors made all data underlying the findings in their manuscript fully available?

Reviewer #1: Yes

Reviewer #2: No

4. Is the manuscript presented in an intelligible fashion and written in standard English?

Reviewer #1: Yes

Reviewer #2: Yes

5. Review Comments to the Author

Reviewer #1: In their study, the authors have examined the effects of sex, diet and antibiotic treatment to the fecal microbiome and hippocampal glial morphology in mice. One of the main problems of the manuscript is that – even though the authors state several times the the gut-brain-axis might play a crucial role (Abstract, line 21, and line 78 in Intro) – it was not examined.

Here are my specific questions and remarks:

• The Abstract is very basic, it does not say which animals were used. Which antibiotics, for how long…..

• Line 42: The CDC reference should be added as a reference.

• Line 59: Please describe here whether the relative abundance of Firmicutes and Bacteroidetes increases or decreases due to high fat diet.

• Line 83: The authors describe the use of antibiotics as an aim without writing a word about them in the Introduction.

• The last two sentences of the introduction (line 84-87) should be deleted. Theyx do not belong to the Introduction.

• Line 106: I think that “..to a subgroup of mice” is confusing. Does this form a new group? Or is this just the antibiotic group. If the second one is true, then the statement could be removed.

• Line 108: How was food consumption measured? Metabolic cages or weighing the food?

• Line 116: How were the fecal samples stored until analysis?

• Line 116: It would be helpful to add the information when the last day of study was. Same for line 142.

• Statistical Analysis, line 186 and 187: The authors have used an ANOVA. Were the data normally distributed? With the relatively small number of animals per group a non-parametric comparison would be more appropriate.

• Results, lines 196-198: These are not really results and should be shown in the Methods section.

• Results, line 201: Do you mean the caloric consumption over the whole study period?

• Results, lines 202 and 203: Are the values shown somewhere? Do the authors mean the summarized calories over the whole study? This is unclear to me.

• Results, lines 203-205: This is not a results but a Discussion.

• Results, lines 205-210 and Figure Caption 1A: This is a lot of redundant enumeration. The authors should consider to add some marks the Figure 1A depicting the statistically significant differences. Same for Figure 1B.

• Figure Caption 1C: This is not weight gain but weight. Please correct.

• Figure 1: Are the values shown mean and SD or medians and IQR. How were the data distributed?

• Figure 1A and B: Either use “LF” or “low fat”. And “HF” or “high fat”

• Figure 1C: What I do not understand is that at week 0 the male HF antibiotics group had about 25 grams mean compared to female LF animals with 17 grams? Please clarify. The weight at week 0 should be more or less the same for all groups. This is confusing.

• Microbiome analysis: I would prefer to read to alpha-diversity first, followed by beta-diversity and relative abundances.

• Table 1: I think Table 1 is confusing and would prefer a Figure showing the relative abundances.

• Lines 240-242: Please refer to the respective Figure here. Do the results remain statistically significant (female LF antibiotics vs. all other antibiotic groups) when comparing all 8 groups?

• Many of the changes found in relative abdundances are below 3%. The biological relevance can be questioned and should be discussed.

• Line 276: delete “as we expected”.

• Caption Figure 3: This is alpha-diversity and should be added.

• Lines 286-294: These are rather Methods and should be described there.

• Figure 5: There is no A etc. shown in the Figure.

• The description of Figure 5 in the text and in the caption is redundant and could be shortened. Moreover, in the caption there is no explanation of “*” etc.

• Line 335: How do we see the distinct clustering of the two groups, there are all dots?

• The Discussion does not follow the line of the results. This should be changed.

• There is no conclusion section. Moreover, the authors should include a limitations of the study section.

• The authors have examined the fecal microbiome and not the gut microbiome. This should be changed throughout the manuscript.

Reviewer #2: PONE-D-21-28243

Sex differences in the gut microbiome and hippocampal glial morphology following diet and antibiotic treatment

The authors examined sex differences in the microbiome changes and hippocampal microgliosis and astrogliosis after consumption of HF diet and antibiotic treatment.

Overall, this was an interesting paper and is well written. I believe that this paper has implications for translation to human research and treatments in obesity. However, the discussion needs some substantial rewriting.

Abstract/Introduction:

1. The abstract needs to summarize the findings and implications of the study better.

Overall, the introduction is well written.

2. However, the rationale for sex differences by summarizing past research was underdeveloped and could use a stronger summary.

3. The specifics from prior research summarizing the links between the key variables was missing.

4. The hypotheses were not specific. What do the authors expect to observe based on the prior literature?

Discussion and Conclusion

1. However, it is still not clear the reason behind the opposite effects observed in males and females as a result of diet and antibiotics and hippocampus glial morphology. More discussion of these results needs to take place. These results seem counterintuitive to past research and to the diet*sex results the authors found.

2. There are a few overstatements of the results. Most of the discussion centers around summary of the results.

3. Most importantly, what do these findings mean? How is this important? Discussion around the integrated results is needed to make these findings meaningful.

4. Minor: the images were slightly blurry. The Double Dendrogram image was extremely difficult to read.

6. PLOS authors have the option to publish the peer review history of their article (what does this mean?). If published, this will include your full peer review and any attached files.

Reviewer #1: No

Reviewer #2: No

---

## [Author Response · Author response to Decision Letter 0]

1 Dec 2021

Response to Reviewers:

We would like to thank the reviewers for their time, comments such as “this was an interesting paper and is well written”, and their thorough review in order to improve our manuscript. We have made substantial revisions throughout the paper, as outlined below and found in our track changes document. 

Reviewer 1:

One of the main problems of the manuscript is that – even though the authors state several times the the gut-brain-axis might play a crucial role (Abstract, line 21, and line 78 in Intro) – it was not examined.

We have removed this from the manuscript and have now listed this as a limitation for our study as well as a goal for future studies.

• The Abstract is very basic, it does not say which animals were used. Which antibiotics, for how long…..

We have now added specific information about the animals and treatments to the Abstract.

• Line 42: The CDC reference should be added as a reference.

We have added this reference.

• Line 59: Please describe here whether the relative abundance of Firmicutes and Bacteroidetes increases or decreases due to high fat diet.

We have clarified that Firmicutes is increased and Bacteroidetes is decreased, supported by our citations.

• Line 83: The authors describe the use of antibiotics as an aim without writing a word about them in the Introduction.

We have added a paragraph about the use of antibiotics as well as clarified their use in this study by stating our hypothesis.

• The last two sentences of the introduction (line 84-87) should be deleted. They do not belong to the Introduction.

Thank you. This has been deleted.

• Line 106: I think that “..to a subgroup of mice” is confusing. Does this form a new group? Or is this just the antibiotic group. If the second one is true, then the statement could be removed.

Thank you. This has been deleted.

• Line 108: How was food consumption measured? Metabolic cages or weighing the food?

We have clarified in the text that food consumption was determined by manually weighing the food.

• Line 116: How were the fecal samples stored until analysis?

• Line 116: It would be helpful to add the information when the last day of study was. Same for line 142.

For the two points above, we have added this to the text (underlined portion): Fecal samples were collected using sterile technique on the last day of the study (after 16 weeks of diet administration, at the time of euthanization) and stored in microcentrifuge tubes at -80o C until analysis.

• Statistical Analysis, line 186 and 187: The authors have used an ANOVA. Were the data normally distributed? With the relatively small number of animals per group a non-parametric comparison would be more appropriate.

Thank you for drawing our attention to this. We have now conducted a Shapiro-Wilk test for these data. The body weight data were not normally distributed. We performed a log transformation of the data and these data were normally distributed. We re-analyzed the log transformed data with a two-way ANOVA and have adjusted the F and p-values, accordingly. This has been updated in the Results section and clarified in the Methods section.

• Results, lines 196-198: These are not really results and should be shown in the Methods section.

Thank you. We moved this to the Methods section.

• Results, line 201: Do you mean the caloric consumption over the whole study period?

Yes. We have added “over the whole study period” to this sentence for clarity.

• Results, lines 202 and 203: Are the values shown somewhere? Do the authors mean the summarized calories over the whole study? This is unclear to me.

We recognize that this added data may cause confusion. We performed a three-way ANOVA for sex x treatment x time for caloric consumption as well as a two-way ANOVA for average food consumption, for the entire study period. We have removed a few sentences in order to improve clarity and only deliver the essential information.

• Results, lines 203-205: This is not a results but a Discussion.

Thank you. We have moved this to the Discussion.

• Results, lines 205-210 and Figure Caption 1A: This is a lot of redundant enumeration. The authors should consider to add some marks the Figure 1A depicting the statistically significant differences. Same for Figure 1B.

We have edited the caption as well as Figures 1A and 1B.

• Figure Caption 1C: This is not weight gain but weight. Please correct.

Thank you for drawing our attention to this error. We have corrected it.

• Figure 1: Are the values shown mean and SD or medians and IQR. How were the data distributed?

We have added that the data reported are mean +/- SEM. We have now clarified which data are normally distributed and any corrections in the Methods section.

• Figure 1A and B: Either use “LF” or “low fat”. And “HF” or “high fat”

We have corrected this error.

• Figure 1C: What I do not understand is that at week 0 the male HF antibiotics group had about 25 grams mean compared to female LF animals with 17 grams? Please clarify. The weight at week 0 should be more or less the same for all groups. This is confusing.

This is typical for males vs. females at 2 months of age, in our previous studies as well as studies from other laboratories. Female mice tend to be smaller than male mice https://www.jax.org/jax-mice-and-services/strain-data-sheet-pages/body-weight-chart-000664#

We included results for weight gain to show change from week 0 to week 16 (Figure 1B) as well as weekly weights in order to show the trajectory for each group (Figure 1C).

• Microbiome analysis: I would prefer to read to alpha-diversity first, followed by beta-diversity and relative abundances.

We agree with your point and have edited the text (and Figures) to present the data in that order.

• Table 1: I think Table 1 is confusing and would prefer a Figure showing the relative abundances.

We have removed Table 1 and have now included a Figure for the top 14 most abundant genera of bacteria.

• Lines 240-242: Please refer to the respective Figure here. Do the results remain statistically significant (female LF antibiotics vs. all other antibiotic groups) when comparing all 8 groups?

We have edited the text to refer to Figure 2A here. Yes – this statistical analysis was performed including all 8 groups, and revealed statistically significant differences for female LF antibiotics compared to all other antibiotic groups.

• Many of the changes found in relative abdundances are below 3%. The biological relevance can be questioned and should be discussed.

As done by Nagpal et al. (https://www.mdpi.com/1422-0067/21/10/3434/htm#B33-ijms-21-03434), we have now excluded data below 0.5% mean relative abundance. The 0.5% abundance may only be met by one of the eight groups. For any genus that did not meet the 0.5% criteria by any group, we have removed it from the table and analysis. Also, we have now included Figure 5 to display the top 14 most abundant genus of bacteria visually. However, we agree that a next step to this, and in future analyses, the biological relevance of the particular genus and threshold abundance for specific impact on metabolite production, etc. will be important. We aim for these data to inform future studies.

• Line 276: delete “as we expected”.

This has been deleted.

• Caption Figure 3: This is alpha-diversity and should be added.

‘Alpha diversity’ has been added.

• Lines 286-294: These are rather Methods and should be described there.

Thank you. We moved this to the Methods section.

• Figure 5: There is no A etc. shown in the Figure.

We have edited the Figure to include A-E.

• The description of Figure 5 in the text and in the caption is redundant and could be shortened. Moreover, in the caption there is no explanation of “*” etc.

We edited the caption to remove redundancy and have included explanation for the asterisks.

• Line 335: How do we see the distinct clustering of the two groups, there are all dots?

We have edited this caption and now refer to “results with low Shannon diversity for antibiotic treated vs. higher Shannon diversity for mice that received normal drinking water”

• The Discussion does not follow the line of the results. This should be changed.

We agree with this point and have edited the text and flow of the Discussion.

• There is no conclusion section. Moreover, the authors should include a limitations of the study section.

We have added a paragraph about limitations and a Conclusion section.

• The authors have examined the fecal microbiome and not the gut microbiome. This should be changed throughout the manuscript.

Thank you for drawing our attention to this error. We have updated our reference to our analyses from “gut microbiome” to “fecal microbiome”. We do still refer to ‘gut microbiome’ generally in the Introduction and Discussion, but whenever we refer to our analyses and results, we state ‘fecal microbiome’. 

Reviewer #2: PONE-D-21-28243

Abstract/Introduction:

1. The abstract needs to summarize the findings and implications of the study better.

We have edited the Abstract to include more details for Methods and Results.

2. Overall, the introduction is well written. However, the rationale for sex differences by summarizing past research was underdeveloped and could use a stronger summary.

3. The specifics from prior research summarizing the links between the key variables was missing.

4. The hypotheses were not specific. What do the authors expect to observe based on the prior literature?

Thank you. For points 2-4, we have edited the Introduction. We now include more information about the use of antibiotics, established sex differences, and a specific hypothesis.

Discussion and Conclusion

1. However, it is still not clear the reason behind the opposite effects observed in males and females as a result of diet and antibiotics and hippocampus glial morphology. More discussion of these results needs to take place. These results seem counterintuitive to past research and to the diet*sex results the authors found.

2. There are a few overstatements of the results. Most of the discussion centers around summary of the results.

3. Most importantly, what do these findings mean? How is this important? Discussion around the integrated results is needed to make these findings meaningful.

For points 1-3, we have made significant changes to the Discussion section. We now discuss previous studies and how our results agree/contradict with those studies. We also now include discussion on what the difference in microglial morphology could be caused by (metabolic factors). Please see our track changes document for all of these changes. 

4. Minor: the images were slightly blurry. The Double Dendrogram image was extremely difficult to read.

We have uploaded new images; they meet the suggested resolution as indicated by the journal so we hope that they no longer appear blurry.

---

## [Decision Letter · Decision Letter 1]

2 Feb 2022

PONE-D-21-28243R1Sex differences in the fecal microbiome and hippocampal glial morphology following diet and antibiotic treatmentPLOS ONE

Dear Dr. Freeman,

Thank you for submitting your manuscript to PLOS ONE. After careful consideration, we feel that it has merit but does not fully meet PLOS ONE’s publication criteria as it currently stands. Therefore, we invite you to submit a revised version of the manuscript that addresses the points raised during the review process.

We look forward to receiving your revised manuscript.

Kind regards,

Christopher Staley, Ph.D.

Academic Editor

PLOS ONE

Journal Requirements:

Additional Editor Comments (if provided):

The authors have addressed the reviewers' comments well, but minor clarifications and additions are still recommended.

Reviewers' comments:

Reviewer's Responses to Questions

**Comments to the Author**

1. If the authors have adequately addressed your comments raised in a previous round of review and you feel that this manuscript is now acceptable for publication, you may indicate that here to bypass the “Comments to the Author” section, enter your conflict of interest statement in the “Confidential to Editor” section, and submit your "Accept" recommendation.

Reviewer #1: (No Response)

Reviewer #3: (No Response)

2. Is the manuscript technically sound, and do the data support the conclusions?

Reviewer #1: Yes

Reviewer #3: Yes

3. Has the statistical analysis been performed appropriately and rigorously? 

Reviewer #1: Yes

Reviewer #3: Yes

4. Have the authors made all data underlying the findings in their manuscript fully available?

Reviewer #1: Yes

Reviewer #3: Yes

5. Is the manuscript presented in an intelligible fashion and written in standard English?

Reviewer #1: Yes

Reviewer #3: Yes

6. Review Comments to the Author

Reviewer #1: The authors have answered the questions and concerns and the manuscript has significantly improved. However, a couple of issues remain:

• Abstract: The study aims should be presented prior to the methods.

• Introduction: The newly added paragraph about antibiotics (lines 89-97) rather reads like a Discussion. Consider to rephrase it and please move the sentence from line 94-97 to the Discussion section.

• For analysis of microglial cells and astrocytes (Methods section): Were there already other papers doing the same thing. If yes, references could be added.

• Figure 2: Would it be possible to add markers of statistically significant differences? This would make it easier for the reader.

• Figure 5: Please write the names with a capital letter first (Lactococcus etc.). Moreover, in lines 317-320 some mentioned genera are not found in the Figure (such as Lactobacillus). Please clarify.

• Legend Figure 6: There is no legend for F.

• Sentence line 401: “Overall, Male and Female LF Antibiotics revealed decreased GFAP density”…this is a bit misleading and therefore “…reaching statistical significance in the female LF antibiotics compared to the female LF group.” should be added.

• The sentence in line 403-404 can be removed.

• The Discussion should start with a short paragraph describing the main findings of the study.

Reviewer #3: (No Response)

7. PLOS authors have the option to publish the peer review history of their article (what does this mean?). If published, this will include your full peer review and any attached files.

Reviewer #1: No

Reviewer #3: No

---

## [Author Response · Author response to Decision Letter 1]

17 Feb 2022

Response to Reviewers:

Thank you for your time and thorough review. Please find our response to the specific comments, below. We have also included a track changes document.

Reviewer 1: The authors have answered the questions and concerns and the manuscript has significantly improved. However, a couple issues remain:

1) Abstract: The study aims should be presented prior to the methods.

This has been fixed.

2) Introduction: The newly added paragraph about antibiotics (lines 89-97) rather reads like a Discussion. Consider to rephrase it and please move the sentence from line 94-97 to the Discussion section.

We have moved line 94-97 to the Discussion section. We find that the part of this paragraph that remains serves as some background/introduction for the use of antibiotics for our study and would prefer to leave that information in the Introduction.

3) For analysis of microglial cells and astrocytes (Methods section): Were there already other papers doing the same thing. If yes, references could be add.

Yes. Thank you for pointing this out. We have now included a few different references in these sections.

4) Figure 2: Would it be possible to add markers of statistically significant differences? This would make it easier for the reader.

We now include indication of statistically significant differences for Figure 2.

5) Figure 5: Please write the names with a capital letter first (Lactococcus etc.). Moreover, in lines 317-320 some mentioned genera are not found in the Figure (such as Lactobacillus). Please clarify.

We have fixed the legend to include a capital letter for the genera. And, we have removed some of the sentences in the paragraph before Figure 5 in order to improve clarity. Originally, we were referring to the genera of bacteria that did not reveal a statistically significant difference between antibiotic-treated and normal drinking water. However, the graph, as you point out, does not clearly show this due to the very high relative abundance of Lactococcus for all antibiotic-treated groups. Therefore, we will keep it to a simple statement and let the Figure as well as the table that follows reveal the rest of the information.

6) Legend Figure 6: There is no legend for F.

Thank you for catching this error. This has been fixed.

7) Sentence line 401: “Overall, Male and Female LF Antibiotics revealed decreased GFAP density”….this is a bit misleading and therefore “…reaching statistical significance in the female LF group.” should be added.

We have revised this sentence and now include this clarification.

8) The sentence in line 403-404 can be removed.

We have removed this sentence.

9) The Discussion should start with a short paragraph describing the main findings of the study.

We have a short paragraph describing the main findings of the study in the Conclusion section that was added. In order to avoid being redundant, we would prefer to keep the Discussion and Conclusion section as they are.

---

## [Editor Report · Decision Letter 2]

9 Mar 2022

Sex differences in the fecal microbiome and hippocampal glial morphology following diet and antibiotic treatment

PONE-D-21-28243R2

Dear Dr. Freeman,

We’re pleased to inform you that your manuscript has been judged scientifically suitable for publication and will be formally accepted for publication once it meets all outstanding technical requirements.

Kind regards,

Christopher Staley, Ph.D.

Academic Editor

PLOS ONE
---

## [Editor Report · Acceptance letter]

14 Mar 2022

PONE-D-21-28243R2 

Sex differences in the fecal microbiome and hippocampal glial morphology following diet and antibiotic treatment 

Dear Dr. Freeman:

I'm pleased to inform you that your manuscript has been deemed suitable for publication in PLOS ONE. Congratulations! Your manuscript is now with our production department. 

Kind regards, 

on behalf of

Dr. Christopher Staley 

Academic Editor

PLOS ONE